# Homobivalent Lamellarin-Like Schiff Bases: In Vitro Evaluation of Their Cancer Cell Cytotoxicity and Multitargeting Anti-Alzheimer’s Disease Potential

**DOI:** 10.3390/molecules26020359

**Published:** 2021-01-12

**Authors:** Alisa A. Nevskaya, Lada V. Anikina, Rosa Purgatorio, Marco Catto, Orazio Nicolotti, Modesto de Candia, Leonardo Pisani, Tatiana N. Borisova, Almira R. Miftyakhova, Aleksey V. Varlamov, Elena Yu. Nevskaya, Roman S. Borisov, Leonid G. Voskressensky, Cosimo D. Altomare

**Affiliations:** 1Organic Chemistry Department, Peoples’ Friendship University of Russia (RUDN University), 6 Miklukho-Maklaya St., 117198 Moscow, Russia; nevskaya.alisa@gmail.com (A.A.N.); tnborisova@mail.ru (T.N.B.); almira244@gmail.com (A.R.M.); avarlamov@sci.pfu.edu.ru (A.V.V.); nevskaya-eyu@rudn.ru (E.Y.N.); borisov@ips.ac.ru (R.S.B.); lvoskressensky@sci.pfu.edu.ru (L.G.V.); 2Institute of Physiologically Active Compounds, Russian Academy of Sciences, 142432 Chernogolovka, Russia; anikina1970@gmail.com; 3Department of Pharmacy-Pharmaceutical Sciences, University of Bari Aldo Moro, Via E. Orabona 4, 70125 Bari, Italy; rosa.purgatorio@uniba.it (R.P.); marco.catto@uniba.it (M.C.); orazio.nicolotti@uniba.it (O.N.); modesto.decandia@uniba.it (M.d.C.); leonardo.pisani@uniba.it (L.P.); 4Topchiev Institute of Petrochemical Synthesis, Russian Academy of Sciences, 29 Leninskii Prosp., 119991 Moscow, Russia; 5Lomonosov North (Arctic) Federal University, Severnaya Dvina Emb. 17, 163002 Arkhangelsk, Russia

**Keywords:** acetylcholinesterase inhibitors, β-amyloid aggregation, anti-Alzheimer’s disease agents, cytotoxicity, pyrrolo[2,1-*a*]isoquinolines

## Abstract

Marine alkaloids belonging to the lamellarins family, which incorporate a 5,6-dihydro-1-phenylpyrrolo[2,1-*a*]isoquinoline (DHPPIQ) moiety, possess various biological activities, spanning from antiviral and antibiotic activities to cytotoxicity against tumor cells and the reversal of multidrug resistance. Expanding a series of previously reported imino adducts of DHPPIQ 2-carbaldehyde, novel aliphatic and aromatic Schiff bases were synthesized and evaluated herein for their cytotoxicity in five diverse tumor cell lines. Most of the newly synthesized compounds were found noncytotoxic in the low micromolar range (<30 μM). Based on a Multi-fingerprint Similarity Search aLgorithm (MuSSeL), mainly conceived for making protein drug target prediction, some DHPPIQ derivatives, especially bis-DHPPIQ Schiff bases linked by a phenylene bridge, were prioritized as potential hits addressing Alzheimer’s disease-related target proteins, such as cholinesterases (ChEs) and monoamine oxidases (MAOs). In agreement with MuSSeL predictions, homobivalent *para*-phenylene DHPPIQ Schiff base **14** exhibited a noncompetitive/mixed inhibition of human acetylcholinesterase (AChE) with *K*_i_ in the low micromolar range (4.69 μM). Interestingly, besides a certain inhibition of MAO A (50% inhibition of the cell population growth (IC_50_) = 12 μM), the bis-DHPPIQ **14** showed a good inhibitory activity on self-induced β-amyloid (Aβ)_1–40_ aggregation (IC_50_ = 13 μM), which resulted 3.5-fold stronger than the respective mono-DHPPIQ Schiff base **9**.

## 1. Introduction

Pyrrolo[2,1-*a*]isoquinoline is the azaheterocyclic core structure of several alkaloids (e.g., crispines, trolline and lamellarins) endowed with diverse biological activities, including anticancer, antiviral and antibacterial activities [1]. Marine alkaloids incorporating a 5,6-dihydro-1-phenylpyrrolo[2,1-*a*]isoquinoline (henceforth referred to as DHPPIQ) moiety into their structure, like type Ia (saturated) lamellarins (Figure 1), showed cytotoxicity to tumor cells and inhibition of P-glycoprotein (P-gp)-mediated multidrug resistance (MDR), some of them being more potent as P-gp inhibitors than the well-known verapamil [2].

DHPPIQ 2-carbaldehydes [3] and related carbonyl adducts [4] (structure I with major points of diversification in Figure 1), mostly Schiff bases, were recently synthesized by some of us and biologically evaluated. Several DHPPIQ derivatives showed high inhibitory potency toward P-gp, attaining sub-micromolar 50% inhibition of the cell population growth (IC_50_) values, and the ability to reverse in vitro P-gp-mediated resistance in doxorubicin-resistant tumor cells [4].

In this study, a number of novel Schiff bases, including homobivalent derivatives, were prepared by condensation of DHPPIQ 2-carbaldehydes with aromatic (*p*-anisidine and *p*-phenylenediamine) and aliphatic amines (e.g., 1,2-ethylenediamine and 1,3-propylenediamine) and their cytotoxicity evaluated in vitro in a panel of five tumor cell lines. With the aim of deriving a spectrum of probable protein targets for the newly synthesized DHPPIQ Schiff bases, we used MuSSeL (Multi-fingerprint Similarity Search aLgorithm), a web server recently developed by some of us [5,6]. Based on the prioritization suggested by the similarity algorithm, the interactions of DHPPIQ 2-aldehyde adducts with cholinesterases (ChEs) and other Alzheimer’s disease (AD)-related targets (i.e., β-amyloid aggregation and monoamine oxidases) were evaluated, which led us to identify some DHPPIQ derivatives as hits of prospective multitarget-directed ligands (MTDLs) for this fatal neurodegenerative disorder.

## 2. Results and Discussion

### 2.1. Chemistry

According to an affordable and effective procedure previously reported by some of us [3], the synthesis of DHPPIQ 2-aldehydes **1–6** was accomplished through a domino reaction of 1-aroyl-3,4-dihydroisoquinolines with α,β-unsaturated aldehydes. The aldehyde derivatives **1–6** were condensed with 4-anisidine in anhydrous toluene to afford the aromatic Schiff bases **7–12** (Scheme 1). The reaction of aldehydes **1**, **3** and **4** with 1,4-phenylenediamine (2:1 molar ratio) in anhydrous toluene led to the formation of homobivalent aromatic Schiff base adducts **13–15**, subsequently transformed into HCl salts to increase their solubility in aqueous buffers. The condensation of **1** with hydrazine or ethylenediamine in anhydrous toluene yielded the homobivalent aliphatic Schiff bases **18** and **19**, respectively. The preparation of compound **16** was accomplished by reacting **1** with 3-(dimethylamino)-1-propylamine in anhydrous benzene. The diamino derivative **17** was prepared in high yield (82%) through a one-pot reaction of **1** with *N*,*N*′-dimethylethylenediamine in acetonitrile, followed by hydrogenation of the iminium hydroxide in methanol, using NaBH_4_ as the reducing agent. The aqueous solubility of **16** and **17** was increased by their transformation into the corresponding hydrochloride (**16a**) and fumarate (**17a**) salts, respectively.

Overall, the optimized synthetic procedures enabled us to prepare novel aromatic and aliphatic Schiff bases of DHPPIQ 2-aldheydes, including some homobivalent derivatives, from fair to high yields.

### 2.2. Cytotoxicity Evaluation

The cytotoxicity of the newly synthesized Schiff bases was evaluated in vitro in a panel of cultured tumor cells, assessing cell viability at scalar concentrations of the test compounds (≤500 μM) by MTT (i.e., 3-(4,5-dimethylthiazol-2-yl)-2,5-diphenyltetrazolium bromide) [7] and Alamar Blue assays [8], using doxorubicin as the positive controls. Most of the compounds were tested against RD (rhabdomyosarcoma), HCT116 (intestinal carcinoma), HeLa (adenocarcinoma of the cervix uterus) and A549 (lung adenocarcinoma) cell lines, whereas five representative compounds of the subsets were tested also against K562 (chronic myelogenous leukemia) tumor cells. The concentrations causing 50% inhibition of the cell population growth (IC_50_ values, µM) are summarized in Table 1.

Together with the cytotoxicity data of the new DHPPIQ Schiff bases (**7–19**), the IC_50_ values of the previously synthesized parent aldehydes (**3–6**) are listed in Table 1; the cytotoxicity of **1** (R^1^ = Et, R^2^ = R^3^ = OEt and R^4^ = H) and **2** (R^1^ = Et, R^2^ = R^3^ = OEt and R^4^ = Ph), whose IC_50_ values were comparable to those of the pairs **3**/**4** and **5**/**6**, were already reported [3].

The newly synthesized Schiff bases did not show appreciable cytotoxicity in the low micromolar range of concentrations (<30 μM). The amino derivative **17a** proved to be more cytotoxic than the closest aliphatic Schiff bases. As a general structure–cytotoxicity correlation trend, it can be noticed that (i) Schiff bases are generally less cytotoxic than the parent aldehydes, (ii) the DHPPIQ adducts bearing a phenyl as R^4^ at C3 are generally less cytotoxic than the corresponding unsubstituted (R^4^ = H) compounds and (iii) homobivalent DHPPIQ derivatives are markedly less cytotoxic than the corresponding mono-DHPPIQ Schiff bases.

### 2.3. Target Protein Prediction by Similarity Search

By employing the recently developed MuSSeL web server [5,6], DHPPIQ and related Schiff base scaffolds representative of compounds **7–12**, **13–15** and **16–19** (Scheme 1) were screened in the attempt to generate a spectrum of putative protein drug targets and to prioritize in vitro biological studies on the new compounds. Interestingly, MuSSeL characterizes DHPPIQ as a privileged structure sharing both evident and latent molecular frameworks with many known bioactive compounds. Based on multi-fingerprint similarity analyses, the DHPPIQ nucleus was found as a potential hit biasing sixteen relevant protein drug targets (Appendix A), including acetylcholinesterase (AChE) from electric eel (*ee*) scored as the sixth target (id CHEMBL4078) and ChE from equine serum (*eq*) scored as the ninth target (id CHEMBL5763).

MuSSeL was further queried with three scaffolds representative of the Schiff bases synthesized in this study, and in all cases, human and heterologous ChEs (both AChE and butyrylcholinesterase (BChE)) were predicted among the most relevant protein targets. In particular, the prototypical twin structure (Appendix A) of the bis-DHPPIQ Schiff base with 1,4-phenylenediamine was paired to *ee*AChE (id CHEMBL4078), as the first target, and to horse ChE (id CHEMBL5763) and *eq*BChE (id CHEMBL5077) as the second and third targets, respectively. Other four human and heterologous ChE isoforms were identified among the prioritized targets for this homobivalent structure, and in this case, human MAO A and MAO B were classified as the sixth and seventh targets.

The ability of small organic molecules to evade or inhibit P-gp may favor their blood–brain barrier (BBB) crossing and distribution into the brain [9]. Taking this notion into account, the good P-gp inhibition capacity demonstrated for very similar DHPPIQ-containing derivatives [3], combined with the outputs of the MuSSeL web server, prompted us to experimentally investigate the interference of the newly synthesized DHPPIQ Schiff bases with three AD-related targets, namely ChE isoforms (AChE and BChE), a β-amyloid aggregation that is a hallmark of AD occurrence and MAO A and B.

### 2.4. Inhibition of AD-Related Targets

Albeit ChE inhibitors, whose rationale against AD lie on the “old” cholinergic theory, provide only limited or palliative care, AChE and BChE inhibitors still remain among the few currently available treatments of AD patients [10,11]. Most of the new DHPPIQ Schiff bases, along with some parent aldehydes (**3–5**), were firstly tested as inhibitors of human AChE and BChE, using the Ellman’s method according to the reported procedure [12] at 10-μM concentration. Molecules achieving at least 60% inhibition at 10 μM were tested at lower scalar concentrations, and their half-maximal inhibitory concentrations (IC_50_ values) were estimated from the best-fitting inhibition–concentration curves. The ChE inhibition data are reported in Table 2.

All the assayed compounds showed selectivity toward AChE. The inhibitory potencies of the homobivalent DHPPIQ aromatic Schiff bases **13a** and **14** (IC_50_ = 13 and 7.3 μM, respectively) resulted about two to three times stronger than those of the corresponding mono-DHPPIQ Schiff bases with *p*-anisidine **7** and **9**. The 3-phenyl DHPPIQ 2-carbadehyde **4** was the only parent aldehyde achieving a finite IC_50_ < 10 μM. The homobivalent DHPPIQ Schiff base with 1,4-phenylenediamine (**14**) proved to be more active than the respective bis-adduct with 1,2-ethylenediamine (**19**). The fumarate salt of the bis-amino derivative **17a** proved to inhibit AChE, with the IC_50_ (16 μM) close to the cytotoxic concentrations.

The complete kinetics showed for **14** and **17a** a noncompetitive/mixed-type inhibition mechanism (Figure 2), with inhibition constant (*K*_i_) values equal to 4.69 ± 0.77 and 13.6 ± 0.10 μM, respectively. The aldehyde derivative **4** also proved to inhibit human AChE with a noncompetitive/mixed-type mechanism (not shown) and *K*_i_ of 3.58 ± 0.04 μM. The most active homobivalent derivative **14** was tenfold less potent than the reference compound galantamine. Its mixed inhibition mechanism on *h*AChE suggests that the DHPPIQ moiety may interact with the peripheral anionic site (PAS) of the enzyme, as suggested by MuSSeL for similar alkaloid structures [13].

Molecules able to inhibit simultaneously AChE and β-amyloid (Aβ) aggregation/neurotoxicity may have potential as drugs for the treatment of AD [14]. Herein, the new DHPPIQ derivatives were evaluated for the inhibitory effects on the self-aggregation of Aβ peptides 1–40 (Aβ_40_), through a test based on the measurement of thioflavin T (ThT) fluorescence [15]. Quercetin, which is a strong inhibitor of in vitro Aβ self-aggregation, was used as a positive control.

Each compound was initially tested at a 100-μM concentration. For compounds showing more than 60% inhibition at 100 μM, IC_50_ values were determined by interpolation of the concentration–response sigmoid curves. Aβ inhibitory data are summarized in Table 2. All the newly synthesized compounds, although less active than quercetin, turned out to be inhibitors of Aβ aggregation of moderate potency at a 100-μM concentration. Interestingly, compound **14** achieves at a low micromolar concentration a dual in vitro pharmacological effect that may result in an enhancement of cholinergic transmission and inhibition of Aβ fibrillization.

Regarding the structure–activity relationships (SARs), the aromatic Schiff bases were found more potent than the aliphatic ones (e.g., **13a** vs. **19**), pointing out that, besides other physicochemical features, the aromatic interactions achieved by the phenylene bridge play a role in breaking protein–protein interactions underpinning Aβ_40_ fibrilization. The homobivalent derivatives (**13a**, **14** and **15a**) proved to be always more potent than the corresponding mono-DHPPIQ aromatic Schiff bases (**7**, **9** and **10**). The most active twin DHPPIQ Schiff base **14** (IC_50_ 13 μM) was found 3.5-fold stronger as an inhibitor of Aβ_40_ aggregation than the respective mono-DHPPIQ derivative **9** (IC_50_ 46 μM). The planarity and aromatic rings in the DHPPIQ moiety could be a key feature for β-sheet intercalation and disruption, while the imino nitrogen in the side chain of the mono-DHPPIQ Schiff bases or in the bridge connecting the two DHPPIQ heads may behave as hydrogen bond (HB) acceptors and dipolar-interacting groups with the peptide backbone [16,17]. The lipophilicity and/or size of the R^1^-R^4^ substituents on the DHPPIQ scaffold also affects the capacity of inhibiting the Aβ_40_ self-aggregation. As a matter of fact, compound **14** (R^1^ = Me, R^2^ = H, R^3^ = Cl and R^4^ = H) turned out a more potent inhibitor than the more lipophilic and bulkier **15a** (R^1^ = Me, R^2^ = H, R^3^ = Cl and R^4^ = Ph) and **13a** (R^1^ = Et, R^2^ = OEt, R^3^ = OEt and R^4^ = H). A similar trend of nonlinear relation of the Aβ_40_ inhibition potency is shown by the mono-DHPPIQ Schiff bases, with the 4′-Cl-phenyl congener **9** sharply more active than the more lipophilic and sterically hindered **10** and slightly more active than **7** (bulkier and more lipophilic) and **11** (less lipophilic). The limited space of the binding site(s) on Aβ_40_ fibrils for the interaction of aromatic/hydrophobic moieties could be a probable explanation of this SAR trend.

The DHPPIQ-containing Schiff bases **9**, **11**, **13a**, **14** and **17a**, as representative of most active AChE and/or Aβ aggregation inhibitors, were tested also on human MAO A and B, using pargyline as the positive control. MAOs were established as relevant targets in neurological disorders: MAO A selective inhibitors are used as antidepressants [18], whereas MAO B selective inhibitors are typically used in the treatment of early symptoms of Parkinson’s disease (PD) [19]. The neuroprotective effects of MAO B inhibitors provide also the rationale for their use in Alzheimer’s disease (AD), associated to their efficacy in decreasing oxidative stress [20]. MAO A and B inhibition data, along with neuronal cytotoxicity expressed as IC_50_ determined by a cell viability assay in SH-SY5Y neuroblastoma cells (camptothecin used as the positive control), are summarized in Table 3. With the exception of **17a**, the tested compounds showed low selectivity toward MAO A. The homobivalent DHPPIQ derivatives **13a** and **14** achieved IC_50_ values toward MAO A at concentrations (ca. 12 μM) significantly lower than the respective IC_50_ values for neuronal toxicity.

## 3. Materials and Methods

### 3.1. Chemistry

Materials and general procedures. All reagents and solvents were purchased from Merck (Darmstadt, Germany), J.T. Baker (Phillipsburg, NJ, USA) or Sigma-Aldrich Chemical Co. (St. Louis, MO, USA) and, unless specified, used without further purification. The melting points (m.p.) of all the compounds were determined on a SMELTING POINT 10 apparatus in open capillaries (Bibby Sterilin Ltd., Stone, UK). IR spectra were recorded on an Infralum FT-801 FTIR spectrometer (ISP SB RAS, Novosibirsk, Russia). The samples were analyzed as KBr disk solids, and the more important frequencies are shown in cm^−1^. ^1^H and ^13^C NMR spectra were recorded in chloroform-*d_3_* (CDCl_3_) or dimethylsulfoxide-*d_6_* (DMSO-d_6_) solutions at 25 °C, with a 600-MHz NMR spectrometer (JEOL Ltd., Tokyo, Japan). Peak positions were given in parts per million (ppm, δ) referenced to the appropriate solvent residual peak, and signal multiplicities were collected by: s (singlet), d (doublet), t (triplet), q (quartet), dd (doublet of doublets), br.s (broad singlet) and m (multiplet). MALDI mass spectra were recorded using a Bruker autoflex speed instrument operating in positive reflectron mode (Bremen, Germany). Elemental analyses were carried out on Euro Vector EA-3000 Elemental Analyzer (Eurovector S.p.A., Milan, Italy) for C, H and N; experimental data agreed to within 0.04% of the theoretical values.

#### 3.1.1. Synthesis of Schiff Bases **7–15**

*p*-Anisidine (for **7–12**) or *p*-phenylenediamine (for **13–15**) (0.88 mmoL) was added to a solution of the corresponding aldehyde derivative (0.44 mmoL) in anhydrous solvent (toluene for **7–12**, MeOH for **13** and EtOH for **14** and **15**). The reaction was carried out in the presence of glacial acetic acid (0.01 mmoL). The mixture was stirred and heated under reflux, and the reaction progress was monitored by thin layer chromatography (TLC). After cooling, the solvent was removed under vacuum, and the residue was recrystallized from EtOAc–hexane to afford compounds **7–12** as a solid. Instead, the isolation of **13–15** was obtained by filtration.

*(E)-1-(1-(3,4-diethoxyphenyl)-8,9-diethoxy-5,6-dihydropyrrolo[2,1-a]isoquinolin-2-yl)-N-(4-methoxyphenyl)methanimine* (**7**). Yellow-green solid, yield 87% (425 mg), M.p. = 144–146 °C. IR cm^−1^: 1602 (C=N). ^1^H NMR (600 MHz, CDCl_3_), *δ*_H_: 1.18 (t, 3H, *J* = 7.1 Hz, OCH_2_CH_3_), 1.37–1.44 (m, 6H, OCH_2_CH_3_), 1.46 (t, 3H, *J* = 7.1 Hz, OCH_2_CH_3_), 3.01 (t, 2H, *J* = 6.3 Hz, 6-CH_2_), 3.60 (q, 2H, *J* = 7.1 Hz, OCH_2_CH_3_), 3.78 (s, 3H, OCH_3_), 3.99–4.08 (m, 4H, OCH_2_CH_3_), 4.10–4.14 (m, 4H, OCH_2_CH_3_, 5-CH_2_), 6.61 (s, 1H, 7-H), 6.69 (s, 1H, 10-H), 6.83 (d, 2H, *J* = 8.3 Hz, Ar-OCH_3_), 6.89–6.95 (m, 3H, CH-Ar), 7.04 (d, 2H, *J* = 8.3 Hz, Ar-OCH_3_), 7.46 (s, 1H, 3-H), 8.15 (s, 1H, CH=N). ^13^C NMR (150 MHz, CDCl_3_), *δ*_C_: 14.7, 14.9 (2C), 15.0, 15.4, 29.3, 45.2, 55.6, 63.9, 64.5, 64.8, 66.0, 109.1, 112.4, 113.8, 114.3 (2C), 116.1, 120.1, 121.8, 122.0 (2C), 122.1, 123.4, 123.9, 126.4, 127.8, 146.2, 146.9, 147.3, 147.9, 148.9, 154.1, 157.5. MALDI, *m/z*: 555 [M+H]^+^. Anal. calc. for C_34_H_38_N_2_O_5_, %: C, 73.64; H, 6.99; N, 5.05 found, %: C, 73.42; H, 7.31; N, 5.19.

*(E)-1-(1-(3,4-diethoxyphenyl)-8,9-diethoxy-3-phenyl-5,6-dihydropyrrolo[2,1-a]isoquinolin-2-yl)-N-(4-methoxyphenyl)methanimine* (**8**). Beige solid, yield 85% (472 mg), M.p. = 93–95 °C. IR cm^−1^: 1606 (C=N). ^1^H NMR (600 MHz, CDCl_3_), δ_H_: 1.20 (t, 3H, *J* = 7.1 Hz, OCH_2_CH_3_), 1.39 (t, 3H, *J* = 7.1 Hz, OCH_2_CH_3_), 1.43 (t, 3H, *J* = 7.1 Hz, OCH_2_CH_3_), 1.47 (t, 3H, *J* = 7.1 Hz, OCH_2_CH_3_), 2.92 (t, 2H, *J* = 6.5 Hz, 6-CH_2_), 3.61 (q, 2H, *J* = 7.1 Hz, OCH_2_CH_3_), 3.73 (s, 3H, OCH_3_), 3.97 (t, 2H, *J* = 7.1 Hz, 5-CH_2_), 4.04–4.09 (m, 4H, OCH_2_CH_3_), 4.14 (q, 2H, *J* = 7.1 Hz, OCH_2_CH_3_), 6.68 (s, 1H, 7-H), 6.69 (s, 1H, 10-H), 6.75 (d, 2H, *J* = 8.6 Hz, Ar-OCH_3_), 6.81 (d, 2H, *J* = 8.6 Hz, Ar-OCH_3_), 6.91 (d, 1H, *J* = 8.1 Hz, CH-Ar), 6.99 (dd, 1H, *J* = 1.5, 8.1 Hz, CH-Ar), 7.07 (d, 1H, *J* = 1.5 Hz, CH-Ar), 7.39–7.41 (m, 1H, CH-Ph), 7.45–7.49 (m, 4H, CH-Ph), 8.22 (s, 1H, CH=N). ^13^C NMR (150 MHz, CDCl_3_), δ_C_: 14.7, 14.9, 15.0 (2C), 29.5, 42.5, 55.5, 55.8, 63.9, 64.5, 64.7 (2C), 109.8, 113.1, 113.6, 114.0 (3C), 114.9, 116.5, 116.7, 119.5, 120.7, 121.8 (3C), 122.1, 123.5, 124.7, 126.8, 128.1, 128.2 (2C), 128.5, 128.8, 130.9, 131.1, 131.2 (2C), 135.2. MALDI, *m/z*: 631 [M+H]^+^. Anal. calc. for C_40_H_42_N_2_O_5_, %: C, 76.19; H, 6.71; N, 4.44; found, %: C, 76.38; H, 6.53; N, 4.54.

*(E)-1-(1-(4-chlorophenyl)-8,9-dimethoxy-5,6-dihydropyrrolo[2,1-a]isoquinolin-2-yl)-N-(4-methoxyphenyl)methanimine* (**9**). Yellow solid, yield 82% (341 mg), M.p. = 153–154 °C. IR cm^−1^: 1617 (C=N). ^1^H NMR (600 MHz, CDCl_3_), *δ*_H_: 3.04 (t, 2H, *J* = 6.2 Hz, 6-CH_2_), 3.40 (s, 3H, OCH_3_), 3.78 (s, 3H, OCH_3_), 3.86 (s, 3H, OCH_3_), 4.11 (t, 2H, *J* = 6.2 Hz, 5-CH_2_), 6.51 (s, 1H, 7-H), 6.70 (s, 1H, 10-H), 6.84 (d, 2H, *J* = 8.8 Hz, Ar-OCH_3_), 7.05 (d, 2H, *J* = 8.8 Hz, Ar-OCH_3_), 7.37–7.41 (m, 4H, Ar-Cl), 7.47 (s, 1H, 3-H), 8.12 (s, 1H, CH=N). ^13^C NMR (150 MHz, CDCl_3_), *δ*_C_: 29.3, 45.2, 55.3, 55.6 (2C), 56.1, 107.4 (2C), 111.4 (2C), 114.4 (3C), 120.3, 120.7 (2C), 121.5, 121.8, 122.0 (3C), 124.3, 126.7, 128.8, 132.5 (2C), 133.1, 133.8. MALDI, *m/z*: 473 [M+H]^+^. Anal. calc. for C_28_H_25_ClN_2_O_3_, %: C, 71.10; H, 5.33; N, 5.92; found, %: C, 71.31; H, 5.18; N, 6.12.

*(E)-1-(1-(4-chlorophenyl)-8,9-dimethoxy-3-phenyl-5,6-dihydropyrrolo[2,1-a]isoquinolin-2-yl)-N-(4-methoxyphenyl)methanimine* (**10**). Yellow solid, yield 76% (367 mg), M.p. = 153–154 °C. IR cm^−1^: 1617 (C=N). ^1^H NMR (600 MHz, CDCl_3_), *δ*_H_: 2.95 (t, 2H, *J* = 6.4 Hz, 6-CH_2_), 3.39 (s, 3H, OCH_3_), 3.73 (s, 3H, OCH_3_), 3.86 (s, 3H, OCH_3_), 3.98 (t, 2H, *J* = 6.4 Hz, 5-CH_2_), 6.50 (s, 1H, 7-H), 6.69 (s, 1H, 10-H), 6.76 (d, 2H, *J* = 8.8 Hz, Ar-OCH_3_), 6.81 (d, 2H, *J* = 8.8 Hz, Ar-OCH_3_), 7.36–7.40 (m, 2H, CH-Ar), 7.40–7.45 (m, 1H, CH-Ar), 7.45–7.51 (m, 6H, CH-Ar), 8.22 (s, 1H, CH=N). ^13^C NMR (150 MHz, CDCl_3_), *δ*_C_: 29.5, 42.5, 55.2, 55.6, 56.0, 106.2, 111.0, 114.1 (2C), 118.9, 119.3, 121.6, 121.8 (2C), 125.1, 127.2, 128.3 (2C), 128.4 (2C), 130.6, 131.1 (2C), 132.5, 132.9 (2C), 135.1, 136.4, 146.3, 147.5, 147.6, 152.7 (2C), 157.3. MALDI, *m/z*: 549 [M+H]^+^. Anal. calc. for C_34_H_29_ClN_2_O_3_, %: C, 74.38; H, 5.28; N, 5.10; found, %: C, 74.59; H, 5.51; N, 5.02.

*(E)-1-(1-(4-fluorophenyl)-8,9-dimethoxy-5,6-dihydropyrrolo[2,1-a]isoquinolin-2-yl)-N-(4-methoxyphenyl)methanimine* (**11**). Yellow solid, yield 84% (337 mg), M.p. 160–162 °C. IR cm^−1^: 1615 (C=N). ^1^H NMR (600 MHz, CDCl_3_), *δ*_H_: 3.04 (t, 2H, *J* = 6.3 Hz, 6-CH_2_), 3.39 (s, 3H, OCH_3_), 3.78 (s, 3H, OCH_3_), 3.85 (s, 3H, OCH_3_), 4.12 (t, 2H, *J* = 6.3 Hz, 5-CH_2_), 6.51 (s, 1H, 7-H), 6.69 (s, 1H, 10-H), 6.85 (d, 2H, *J* = 8.6 Hz, Ar-OCH_3_), 7.05 (d, 2H, *J* = 8.6 Hz, Ar-OCH_3_), 7.12 (t, 2H, *J* = 8.3 Hz, Ar-F), 7.40 (dd, 2H, *J* = 5.5, 8.1 Hz, Ar-F), 7.47 (s, 1H, 3-H), 8.11 (s, 1H, CH=N). ^13^C NMR (150 MHz, CDCl_3_), *δ*_C_: 29.3, 45.2, 55.3, 55.6, 56.1, 107.3, 111.4, 114.4, 115.6 (d, 2C, *J* = 21.7 Hz), 115.7, 120.5, 120.6, 121.6, 122.0, 124.2, 126.6, 131.2 (d, 1C, *J* = 4.3 Hz), 132.7 (d, 2C, *J* = 8.7 Hz), 132.8, 146.1, 147.4, 147.7, 153.5 (2C), 157.6, 161.4 (d, *J* = 245.7, 1C). MALDI, *m/z*: 457 [M+H]^+^. Anal. calc. for C_28_H_25_FN_2_O_3_, %: C, 73.68; H, 5.54; N, 6.13; found, %: C, 73.37; H, 5.28; N, 6.03.

*(E)-1-(1-(4-fluorophenyl)-8,9-dimethoxy-3-phenyl-5,6-dihydropyrrolo[2,1-a]isoquinolin-2-yl)-N-(4-methoxyphenyl)methanimine* (**12**). White solid, yield 75% (351 mg), M.p. = 195–197 °C. IR cm^−1^: 1617 (C=N). ^1^H NMR (600 MHz, CDCl_3_), δ_H_: 2.95 (t, 2H, *J* = 6.6, 6-CH_2_), 3.93 (s, 3H, OCH_3_), 3.74 (s, 3H, OCH_3_), 3.86 (s, 3H, OCH_3_), 3.98 (t, 2H, *J* = 6.6, 5-CH_2_), 6.50 (s, 1H, 7-H), 6.69 (s, 1H, 10-H), 6.76 (d, 2H, *J* = 8.6 Hz, Ar-OCH_3_), 6.82 (d, 2H, *J* = 8.6, Ar-OCH_3_), 7.38 (d, 2H, *J* = 8.1 Hz, CH-Ar), 7.43–7.48 (m, 7H, CH-Ar), 8.22 (s, 1H, CH=N).^13^C NMR (150 MHz, CDCl_3_), δ_C_: 28.9, 42.61, 54.9, 55.7, 56.1, 112.4, 114.7, 114.8 (d, 2C, *J* = 20.2 Hz), 118.3, 118.6, 121.0, 121.8 (2C), 126.4, 127.5, 128.9 (2C), 129.0 (d, 1C, *J* = 4.3 Hz), 130.2, 131.5 (2C), 131.8, 133.5 (d, 2C, *J* = 14.4 Hz), 135.7, 137.1, 135.7, 137.1, 146.5, 147.5, 148.0, 153.0, 157.4 (d, 1C, *J* = 244.2 Hz). MALDI, *m/z*: 533 [M+H]^+^. Anal. calc. for C_34_H_29_FN_2_O_3_, %: C, 76.67; H, 5.49; N, 5.26; found, %: C, 76.48; H, 5.62; N, 5.07.

*(1E,1’E)-N,N’-(1,4-phenylene)bis(1-(1-[3,4-diethoxyphenyl]-8,9-diethoxy-5,6-dihydropyrrolo[2,1-a]isoquinolin-2-yl)methanimine)* (**13**). Yellow solid, yield 70% (597 mg), M.p. = 220–221 °C. IR cm^−1^: 1612 (C=N). ^1^H NMR (400 MHz, DMSO-d_6_), δ_H_: 1.06 (t, 6H, *J* = 7.1 Hz, OCH_2_CH_3_), 1.25 (t, 6H, *J* = 7.1 Hz, OCH_2_CH_3_), 1.29 (t, 6H, *J* = 7.1 Hz, OCH_2_CH_3_), 1.33 (t, 6H, *J* = 7.1 Hz, OCH_2_CH_3_), 2.97 (t, 4H, *J* = 5.8 Hz, 6-CH_2_), 3.50 (q, 4H, *J* = 7.1 Hz, OCH_2_CH_3_), 3.94–4.00 (m, 8H, OCH_2_CH_3_), 4.05 (q, 4H, *J* = 7.1 Hz, OCH_2_CH_3_), 4.12 (t, 4H, *J* = 5.8 Hz, 5-CH_2_), 6.53 (s, 2H, 7-H), 6.84–6.87 (m, 4H, 10-H, CH-Ar), 6.92 (br.s, 2H, CH-Ar), 6.94–6.98 (m, 4H, =N-C_6_H_4_-N=), 7.02 (d, 2H, *J* = 8.1 Hz, CH-Ar), 7.57 (s, 2H, 3-H), 8.06 (s, 2H, CH=N). ^13^C NMR (150 MHz, DMSO-d_6_), δ_C_: 14.7 (2C), 14.9 (2C), 15.0 (2C), 29.2 (2C), 29.3 (2C), 45.2 (2C), 45.5 (2C), 63.9 (2C), 64.5 (2C), 64.8 (2C), 109.1 (2C), 109.4, 113.2, 113.4, 113.8, 115.7, 116.0, 120.0, 120.4, 121.3, 121.7 (2C), 121.9, 122.0, 122.1 (2C), 123.2, 123.4 (2C), 133.9 (2C), 124.2, 124.3, 124.6, 126.4, 126.5, 126.8, 127.3, 127.7, 127.9, 144.2, 146.8, 146.9, 147.3, 147.9, 148.9, 150.1, 152.8, 154.6. MALDI, *m/z*: 971 [M+H]^+^. Anal. calc. for C_60_H_66_N_4_O_8_, %: C, 74.21; H, 6.84; N, 5.78; found, %: C, 74.13; H, 6.96; N, 5.90.

*(1E,1’E)-N,N’-(1,4-phenylene)bis(1-(1-[4-chlorophenyl]-8,9-dimethoxy-5,6-dihydropyrrolo[2,1-a]isoquinolin-2-yl)methanimine)* (**14**). Yellow solid, yield 65% (461 mg), M.p. = 260–261 °C. IR cm^−1^: 1615 (C=N). ^1^H NMR (600 MHz, CDCl_3_), *δ*_H_: 3.03 (t, 4H, *J* = 6.3 Hz, 6-CH_2_), 3.40 (s, 6H, OCH_3_), 3.85 (s, 6H, OCH_3_), 4.12 (t, 4H, *J* = 6.3 Hz, 5-CH_2_), 6.50 (s, 2H, 7-H), 6.69 (s, 2H, 10-H), 7.02–7.07 (m, 4H, =N-C_6_H_4_-N=), 7.35–7.41 (m, 8H, CH-Ar), 7.50 (s, 2H, 3-H), 8.12 (s, 2H, CH=N). ^13^C NMR (150 MHz, DMSO-d_6_), *δ*_C_: 29.1, 29.3, 45.2, 45.4, 55.3 (2C), 56.0 (2C), 107.4, 107.7, 111.4 (2C), 115.7 (2C), 120.0, 120.4, 120.5, 120.7, 121.1, 121.3, 121.6, 121.7 (2C), 122.0, 124.1, 124.3, 124.6, 126.2, 126.7, 126.8, 126.9, 127.7, 128.8 (2C), 132.3, 132.5 (2C), 132.9, 133.2, 133.4, 133.6, 147.6, 147.7, 147.8, 148.0, 150.0, 151.8, 154.0. MALDI, *m/z*: 807 [M+H]^+^. Anal. calc. for C_48_H_40_Cl_2_N_4_O_4_, %: C, 71.37; H, 4.99; N, 6.94; found, %: C, 71.54; H, 4.65; N, 7.09.

*(1E,1’E)-N,N’-(1,4-phenylene)bis(1-(1-[4-chlorophenyl]-8,9-dimethoxy-3-phenyl-5,6-dihydropyrrolo[2,1-a]isoquinolin-2-yl)methanimine)* (**15**). Yellow solid, yield 61% (514 mg), M.p. = 299–300 °C. IR cm^−1^: 1618 (C=N). ^1^H NMR (400 MHz, DMSO-d_6_), δ_H_: 2.96 (t, 4H, *J* = 6.1 Hz, 6-CH_2_), 3.40 (s, 6H, OCH_3_), 3.87 (s, 6H, OCH_3_), 3.99 (t, 4H, *J* = 6.1 Hz, 5-CH_2_), 6.51 (s, 2H, 7-H), 6.70 (s, 2H, 10-H), 6.74–6.76 (m, 4H, =N-C_6_H_4_-N=), 7.35–7.38 (m, 4H, CH-Ar), 7.40–7.49 (m, 14H, CH-Ar), 8.19 (s, 2H, CH=N). ^13^C NMR (150 MHz, DMSO-d_6_), δ_C_: 29.3, 29.5, 42.5, 55.2, 56.0, 108.2 (2C), 111.0 (2C), 115.5 (2C), 118.9 (2C), 119.2 (2C), 120.9 (2C), 121.1 (2C), 121.3, 121.6, 121.9 (2C), 125.1 (2C), 125.2 (2C), 127.2, 127.6, 128.3 (2C), 128.4 (2C), 128.6 (2C), 128.7 (2C), 129.1, 129.3 (2C), 130.4, 131.0 (2C), 131.1 (2C), 132.3 (2C), 132.5, 132.9 (2C), 133.2, 133.8, 134.9, 141.7, 147.5, 147.6, 147.7, 148.0, 151.3, 153.1. MALDI, *m/z*: 959 [M+H]^+^. Anal. calc. for C_60_H_48_Cl_2_N_4_O_4_, %: C, 75.07; H, 5.04; N, 5.84; found, %: C, 74.85; H, 5.21; N, 5.91.

*(E)-3-(([1-(3,4-diethoxyphenyl]-8,9-diethoxy-5,6-dihydropyrrolo[2,1-a]isoquinolin-2-yl)methylene)amino)-N,N-dimethylpropan-1-amine* (**16**). 3-Dimethylaminopropylamine (1.11 mmol) was added to a solution of aldehyde **1** (0.44 mmol) in anhydrous benzene. The mixture was stirred and heated under reflux. The progress of the reaction was monitored by TLC, and after cooling, the solvent was removed under vacuum to afford the desired compound **16** as an oil. Yellow oil, yield 99% (73 mg); IR cm^−1^: 1617 (C=N). ^1^H NMR (600 MHz, CDCl_3_), δ_H_: 1.17 (t, 3H, *J* = 7.3 Hz, OCH_2_CH_3_), 1.38–1.42 (m, 6H, OCH_2_CH_3_), 1.46 (t, 3H, *J* = 7.3 Hz, OCH_2_CH_3_), 1.75–1.78 (m, 2H, =N(CH_2_)_3_N(CH_3_)_2_), 2.18 (s, 6H, =N(CH_2_)_3_N(CH_3_)_2_), 2.26 (t, 2H, *J* = 7.6 Hz, =N(CH_2_)_3_N(CH_3_)_2_), 2.97 (t, 2H, *J* = 6.6 Hz, 6-CH_2_), 3.41 (t, 2H, *J* = 6.6 Hz, 5-CH_2_), 3.58 (q, 2H, *J* = 7.3 Hz, OCH_2_CH_3_), 4.00–4.06 (m, 6H, OCH_2_CH_3_, =N(CH_2_)_3_N(CH_3_)_2_), 4.12 (q, 2H, *J* =7.3 Hz, OCH_2_CH_3_), 6.62 (s, 1H, 7-H), 6.66 (s, 1H, 10-H), 6.88–6.89 (m, 2H, CH-Ar), 6.91 (d, 1H, *J* = 8.6 Hz, CH-Ar), 7.28 (s, 1H, 3-H), 7.95 (s, 1H, CH=N). ^13^C NMR (150 MHz, DMSO-d_6_), δ_C_: 14.7, 14.9 (2C), 15.0, 29.3 (2C), 45.1, 45.6 (2C), 57.8, 60.0, 63.9, 64.5, 64.7, 64.8, 109.1, 113.4, 113.7, 116.1, 119.2, 121.2 (2C), 122.1, 123.3, 123.9, 126.1, 128.1, 146.7, 147.2, 147.7, 148.8, 156.3. MALDI, *m/z*: 534 [M+H]^+^. Anal. calc. for C_32_H_43_N_3_O_4_, %: C, 72.01; H, 8.12; N, 7.87; found, %: C, 72.18; H, 8.29; N, 7.74.

#### 3.1.2. Preparation of Quaternary Salts **13a**, **15a** and **16a**

Concentrated hydrochloric acid was added up to pH 2 to a chloroform solution of the Schiff bases **13**, **15** and **16** to yield compounds **13a, 15a** and **16a**, respectively. The solvent was removed under vacuum, and the residue was crystallized in Et_2_O to afford purified compounds as solids with quantitative yields.

*(1E,1’E)-N,N’-(1,4-phenylene)bis(1-(1-[3,4-diethoxyphenyl]-8,9-diethoxy-5,6-dihydropyrrolo[2,1-a]isoquinolin-2-yl)methanimine)-1,4-diaminium dichloride* (**13a**). Yellow solid, yield 99% (55 mg), M.p. = 240–245 °C. IR cm^−1^: 1602 (C=N). ^1^H NMR (600 MHz, CDCl_3_), δ_H_: 1.19 (t, 6H, *J* = 6.8 Hz, OCH_2_CH_3_), 1.40–1.44 (m, 12H, OCH_2_CH_3_), 1.51 (t, 6H, *J* = 6.8 Hz, OCH_2_CH_3_), 3.00–3.10 (m, 4H, 6-CH_2_), 3.60 (q, 4H, *J* = 6.8 Hz, OCH_2_CH_3_), 4.00 (q, 4H, *J* = 6.8 Hz, OCH_2_CH_3_), 4.06 (q, 4H, *J* = 6.8 Hz, OCH_2_CH_3_), 4.18 (q, 4H, *J* = 6.8 Hz, OCH_2_CH_3_), 4.25–4.35 (m, 4H, 5-CH_2_), 6.64 (s, 2H, 7-H), 6.70 (s, 2H, 10-H), 6.85 (s, 2H, CH-Ar), 6.88 (d, 2H, *J* = 7.8 Hz, CH-Ar), 7.02 (d, 2H, *J* = 7.8 Hz, CH-Ar), 7.75–7.77 (m, 4H, =N-C_6_H_4_-N=), 8.02 (br.s, 2H, 3-H), 9.60 (s, 2H, CH=N). ^13^C NMR (150 MHz, DMSO-d_6_), δ_C_:14.6 (2C), 14.8 (2C), 14.9 (4C), 28.7 (2C), 46.6 (2C), 64.1 (2C), 64.7 (2C), 64.9 (4C), 109.5 (2C), 112.9 (2C), 113.9 (2C), 115.3 (2C), 115.5 (2C), 119.4 (2C), 121.3 (4C), 123.2 (2C), 123.8 (2C), 124.0 (2C), 124.8 (2C), 129.4 (2C), 134.2 (2C), 137.9 (2C), 147.6 (2C), 148.5 (2C), 149.4 (2C), 149.5 (2C), 152.4 (2C). MALDI, *m/z*: 971 [M-HCl-Cl^−^]^+^. Anal. calc. for C_60_H_66_N_4_O_8_ x 2HCl, %: C, 69.03; H, 6.54; N, 5.39; found %: C, 68.79; H, 6.73; N, 5.23.

*(1E,1’E)-N,N’-(1,4-phenylene)bis(1-(1-[4-chlorophenyl]-8,9-dimethoxy-3-phenyl-5,6-dihydropyrrolo[2,1-a]isoquinolin-2-yl)methanimine)-1,4-diaminium dichloride* (**15a**). Yellow solid, yield 99% (54 mg), M.p. = 306–307 °C. IR cm^−1^: 1620 (C=N). ^1^H NMR (400 MHz, DMSO-d_6_), δ_H_: 2.97 (t, 4H, *J* = 6.6 Hz, 6-CH_2_), 3.38 (s, 6H, OCH_3_), 3.85 (s, 6H, OCH_3_), 3.97 (t, 4H, *J* = 6.6 Hz, 5-CH_2_), 6.46 (s, 2H, 7-H), 6.68 (s, 2H, 10-H), 7.25–7.28 (m, 4H, =N-C_6_H_4_-N=), 7.39–7.42 (m, 8H, CH-Ar), 7.45–7.47 (m, 4H, CH-Ph), 7.49–7.51 (m, 6H, CH-Ph), 9.62 (s, 2H, CH=N). ^13^C NMR (150 MHz, DMSO-d_6_), δ_C_: 29.3 (2C), 42.5 (2C), 55.2 (2C), 56.0 (2C), 108.2 (4C), 111.0 (4C), 119.0 (2C), 120.9 (2C), 121.4 (2C), 125.2 (4C), 128.7 (6C), 129.1 (2C), 129.3 (4C), 131.0 (6C), 132.3 (4C), 133.2 (4C), 133.8 (2C), 141.7 (2C), 147.7 (2C), 148.0 (2C). MALDI, *m/z*: 959 [M-HCl-Cl^−^]^+^. Anal. calc. for C_60_H_48_Cl_2_N_4_O_4_ x 2HCl, %: C, 69.77; H, 4.88; N, 5.42; found %: C, 69.88; H, 4.95; N, 5.53.

*(E)-N^1^-((1-[3,4-diethoxyphenyl]-8,9-diethoxy-5,6-dihydropyrrolo[2,1-a]isoquinolin-2-yl)methylene)-N^3^,N^3^-dimethylpropane-1,3-diaminium chloride* (**16a**). Yellow solid, yield 99% (57 mg), M.p. 75–76 °C. IR cm^−1^: 1653 (C=N). ^1^H NMR (600 MHz, DMSO-d_6_), δ_H_: 1.02 (t, 3H, *J* = 6.6 Hz, OCH_2_CH_3_), 1.23–1.27 (m, 6H, OCH_2_CH_3_), 1.31 (t, 3H, *J* = 6.6 Hz, OCH_2_CH_3_), 1.87–1.89 (m, 2H, =N(CH_2_)_3_N(CH_3_)_2_), 2.47 (s, 6H, =N(CH_2_)_3_N(CH_3_)_2_), 2.84 (t, 2H, *J* = 7.1 Hz, 6-CH_2_), 2.89–2.95 (m, 4H, =N(CH_2_)_3_N(CH_3_)_2_), 3.45 (q, 2H, *J* = 6.6 Hz, OCH_2_CH_3_), 3.92–3.97 (m, 4H, OCH_2_CH_3_) 4.02–4.03 (m, 2H, OCH_2_CH_3_), 4.09–4.11 (m, 2H, 5-CH_2_), 6.47 (s, 1H, 7-H), 6.82–6.84 (m, 2H, 10-H, CH-Ar), 6.87 (s, 1H, CH-Ar), 6.97 (d, 1H, *J* = 8.1 Hz, CH-Ar), 7.63 (s, 1H, 3-H), 9.50 (br.s, 1H, CH=N). ^13^C NMR (150 MHz, DMSO-d_6_), δ_C_: 14.7, 14.9 (2C), 15.0, 29.3 (2C), 45.1, 45.6 (2C), 57.8, 60.0, 63.9, 64.5, 64.7, 64.8, 109.1, 113.4, 113.7, 116.1, 119.2, 121.2 (2C), 122.1, 123.3, 123.9, 126.1, 128.1, 146.7, 147.2, 147.7, 148.8, 156.3. MALDI, *m/z*: 534 [M-HCl-Cl^−^]^+^. Anal. calc. For C_32_H_43_N_3_O_4_ x 2HCl, %: C, 71.74; H, 8.47; N, 7.84; found, %: C, 71.64; H, 8.63; N, 7.63.

*N^1^-((1-[3,4-diethoxyphenyl]-8,9-diethoxy-5,6-dihydropyrrolo[2,1-a]isoquinolin-2-yl)methyl)-N^1^,N^2^-dimethylethane-1,2-diamine* (**17**). N,N’-dimethylethylenediamine (1.11 mmol) was added to a solution of aldehyde **1** (0.44 mmol) in acetonitrile. The reaction was carried out with anhydrous MgSO_4_ (2.0 mmol). The mixture was stirred and heated under reflux for 8 h. Then, the solvent was changed to methanol, and NaBH_4_ (3.33 mmol) was added to the mixture portion-wise. The reaction was monitored by TLC. After cooling, the solvent was removed under vacuum, and water was added to the residue and extracted 3 times with EtOAc. The residue was recrystallized from EtOAc–hexane to afford compound **17** as a solid. Beige solid, yield 82% (190 mg); M.p. = 201–202 °C. ^1^H NMR (600 MHz, CDCl_3_), δ_H_: 1.15 (t, 3H, *J* = 7.1 Hz, OCH_2_CH_3_), 1.37–1.41 (m, 6H, OCH_2_CH_3_), 1.45 (t, 3H, *J* = 7.1 Hz, OCH_2_CH_3_), 2.35 (s, 3H, NCH_3_), 2.49 (s, 3H, NCH_3_), 2.95–2.98 (m, 4H, CH_2_N(CH_2_)_2_N), 3.00–3.04 (m, 2H, 6-CH_2_), 3.55 (q, 2H, *J* = 7.1 Hz, OCH_2_CH_3_), 3.77 (br.s., 2H, CH_2_N(CH_2_)_2_N), 3.99–4.05 (m, 6H, OCH_2_CH_3_, 5-CH_2_), 4.09–4.14 (m, 2H, OCH_2_CH_3_), 6.50 (s, 1H, 7-H), 6.66 (s, 1H, 10-H), 6.80–6.85 (m, 2H, CH-Ar), 6.95–6.97 (m, 2H, CH-Ar, 3-H). ^13^C NMR (150 MHz, DMSO-d_6_), δ_C_: 14.5, 14.6, 14.9 (2C), 29.1, 31.6, 33.5, 44.3, 45.0, 45.2, 51.1, 51.3, 63.9, 64.7, 64.9, 109.1 (2C), 113.4, 114.2, 114.3, 116.0, 121.0, 121.3, 122.9, 123.0, 123.3, 124.3, 127.4, 147.2, 148.3, 149.6. MALDI, *m/z*: 522 [M+H]^+^. Anal. calc. for C_31_H_43_N_3_O_4_, %: C, 71.38; H, 8.32; N, 8.04; found, %: C, 71.26; H, 8.15; N, 8.22.

*2-(([1-(3,4-diethoxyphenyl]-8,9-diethoxy-5,6-dihydropyrrolo[2,1-a]isoquinolin-2-yl)methyl)(methyl)amino)-N-methylethan-1-aminium (E)-3-carboxyacrylate* (**17a**). A solution of fumaric acid (0.44 mmol) in 3-mL ethanol was added to the solution of amine **17** (0.39 mmol) in 3 mL of absolute ethanol. The reaction was heated to a boil and then cooled to room temperature. After the addition of 1 mL of ether, the desired compound **17a** crystalized and, then, was isolated by filtration (quantitative yield). Beige solid, yield 99% (246 mg); M.p. = 201–202 °C. ^1^H NMR (400 MHz, DMSO-d_6_), δ_H_: 1.05 (t, 3H, *J* = 6.6 Hz, OCH_2_CH_3_), 1.26–1.29 (m, 6H, OCH_2_CH_3_), 1.33 (t, 3H, *J* = 6.6 Hz, OCH_2_CH_3_), 2.11 (s, 3H, N-CH_3_), 2.44 (s, 3H, N-CH_3_), 2.79–2.81 (m, 2H, 6-CH_2_), 2.88 (t, 2H, *J* = 6.1 Hz, CH_2_N(CH_2_)_2_N), 2.92 (t, 2H, *J* = 6.1 Hz, CH_2_N(CH_2_)_2_N), 3.23 (s, 2H, CH_2_N(CH_2_)_2_N), 3.47 (q, 2H, *J* = 6.6 Hz, OCH_2_CH_3_), 3.90–3.94 (m, 6H, OCH_2_CH_3_, 5-CH_2_), 4.03 (q, 2H, *J* = 6.6 Hz, OCH_2_CH_3_), 6.44–6.46 (m, 4H, 7-H, 10-H, HO_2_C-(CH)_2_-CO_2_H), 6.77–6.81 (m, 3H, CH-Ar, 3-H), 6.95 (d, 1H, *J* = 8.1 Hz, CH-Ar). ^13^C NMR (150 MHz, DMSO-d_6_), δ_C_: 15.1, 15.3 (2C), 15.4 (2C), 29.1, 32.9, 42.0, 44.4, 46.0, 52.1, 52.5, 63.6 (2C), 64.3, 64.4, 64.5, 65.4, 109.0, 114.2, 114.6, 116.6, 118.4, 120.6, 120.8, 123.4, 124.7, 125.0, 129.6, 135.5, 146.5, 146.7, 147.5, 148.7, 168.3. MALDI, *m/z*: 522 [M-HOOCCHCHCOO^−^]^+^. Anal. calc. for C_31_H_43_N_3_O_4_ x C_4_O_4_H_2_, %: C, 65.91; H, 7.43; N, 6.59; found, %: C, 65.72; H, 7.24; N, 6.43.

#### 3.1.3. Synthesis of Compounds **18** and **19**

Hydrazine or ethan-1,2-diamine (0.67 mmol) and MgSO_4_ (0.67 mmol) were added to a solution of aldehyde **1** (0.44 mmol) in anhydrous toluene. The mixture was stirred and heated under reflux. The reaction was monitored by TLC, and after cooling, the solvent was removed under vacuum. The residue was recrystallized from EtOAc–hexane to afford compounds **18** and **19** as solids.

*(1E,2E)-1,2-bis((1-[3,4-diethoxyphenyl]-8,9-diethoxy-5,6-dihydropyrrolo[2,1-a]isoquinolin-2-yl)methylene)hydrazine* (**18**). Beige solid, yield 73% (285 mg); M.p. = 230–232 °C. IR cm^−1^: 1619 (C=N). ^1^H NMR (600 MHz, CDCl_3_), *δ*_H_: 1.16 (t, 6H, *J* = 7.0 Hz, OCH_2_CH_3_), 1.38–1.42 (m, 12H, OCH_2_CH_3_), 1.47 (t, 6H, *J* = 7.0 Hz, OCH_2_CH_3_), 2.98 (t, 4H, *J* = 6.1 Hz, 6-CH_2_), 3.58 (q, 4H, *J* = 7.0 Hz, OCH_2_CH_3_), 4.0–4.07 (m, 12H, OCH_2_CH_3_, 5-CH_2_), 4.13 (q, 4H, *J* = 7.0 Hz, OCH_2_CH_3_), 6.63 (s, 2H, 7-H), 6.66 (s, 2H, 10-H), 6.87–6.90 (m, 4H, CH-Ar), 6.90–6.94 (m, 2H, CH-Ar), 7.31 (s, 2H, 3-H), 8.32 (s, 2H, CH=N). ^13^C NMR (150 MHz, DMSO-d_6_), *δ*_C_: 14.6 (4C), 15.0 (4C), 29.3 (2C), 45.1 (2C), 63.9 (2C), 64.6 (2C), 64.8 (2C), 65.0 (2C), 109.1 (2C), 113.5 (2C), 114.2 (2C), 116.2 (2C), 119.1 (2C), 119.3 (2C), 121.8 (2C), 122.1 (2C), 123.5 (2C), 123.8 (2C), 126.5 (2C), 128.1 (2C), 146.8 (2C), 147.3 (2C), 147.9 (2C), 149.1 (2C), 155.4 (2C). MALDI, *m/z*: 895 [M+H]^+^. Anal. calc. for C_54_H_62_N_4_O_8_, %: C, 72.46; H, 6.98; N, 6.26; found, %: C, 72.71; H, 6.83; N, 6.40.

*(1E,1’E)-N,N’-(ethane-1,2-diyl)bis(1-(1-(3,4-diethoxyphenyl)-8,9-diethoxy-5,6-dihydropyrrolo[2,1-a]isoquinolin-2-yl)methanimine)* (**19**). Beige solid, yield 65% (256 mg), M.p. = 205–206 °C; IR cm^−1^: 1635 (C=N). ^1^H NMR (600 MHz, CDCl_3_), δ_H_: 1.16 (t, 6H, *J* = 7.1 Hz, OCH_2_CH_3_), 1.36 (t, 6H, *J* = 7.1 Hz, OCH_2_CH_3_), 1.41–1.45 (m, 12H, OCH_2_CH_3_), 2.98 (t, 4H, *J* = 6.6 Hz, 6-CH_2_), 3.57 (q, 4H, *J* = 7.1 Hz, OCH_2_CH_3_), 3.61–3.64 (m, 4H, N(CH_2_)_2_N), 3.97 (q, 4H, *J* = 7.1 Hz, OCH_2_CH_3_), 4.02–4.07 (m, 8H, OCH_2_CH_3_, 5-CH_2_), 4.10 (q, 4H, *J* = 7.1 Hz, OCH_2_CH_3_), 6.61 (s, 2H, 7-H), 6.66 (s, 2H, 10-H), 6.84–6.89 (m, 6H, CH-Ar), 7.24 (s, 2H, 3-H), 7.97 (s, 2H, CH=N). ^13^C NMR (150 MHz, DMSO-d_6_), δ_C_: 15.0 (4C), 15.2 (4C), 15.3 (8C), 22.6 (2C), 28.9 (2C), 31.5 (2C), 62.2, 63.7 (2C), 64.3 (4C), 64.5 (2C), 109.2 (2C), 114.1, 114.3, 116.4, 120.6 (2C), 120.8 (2C), 121.0 (2C), 121.8, 123.4, 125.1, 125.7, 128.1, 146.8 (2C), 146.9 (2C), 147.7, 148.7 (2C), 156.3 (2C). MALDI, *m/z*: 923 [M+H]^+^. Anal. calc. for C_56_H_66_N_4_O_8_, %: C, 72.87; H, 7.23; N, 6.08; found, %: C, 72.70; H, 7.41; N, 6.31.

### 3.2. Chemoinformatics and Computational Chemistry

The Multi-fingerprint Similarity Search aLgorithm (MuSSeL) is released as a ligand-based predictive web server to find putative protein drug targets of new conceived small molecules or to repurpose existing bioactive compounds [5,6]. Predictions are computed by screening a collection, including 611,333 small molecules provided with high-quality experimental bioactivity data covering 3357 protein drug targets, which were rationally selected from the latest release of ChEMBLdb (version 25, March 2019) [21]. In particular, MuSSeL makes use of a pool of 13 selected molecular fingerprints to predict therapeutically relevant protein drug targets based on a consensus scheme for a given user query. Notably, MuSSeL performances benefit from an object-relational database management system based on PostgreSQL. In this respect, the real-life effectiveness of MuSSeL was challenged by predicting a pool of 36 external bioactive compounds published in the Journal of Medicinal Chemistry from October to December 2018. Upon the request of a free license, MuSSeL is publicly available at http://mussel.uniba.it:5000/. The platform allows interested users to screen single or even multiple queries at a time, as normally requested in reverse-screening campaigns, which are part of modern drug discovery pipelines. Noteworthy, our multi-fingerprint search algorithm proved successful also for the prediction of acute oral toxicity [22].

### 3.3. Cell Viability Assays

#### 3.3.1. Cell Lines

Human cell cultures A549 (ATCC^®^ CCL-185™, lung carcinoma), HCT116 (ATCC^®^ CCL-247™, intestinal carcinoma), RD (ATCC^®^ CCL-136™, rhabdomyosarcoma), SH-SY5Y (ATCC^®^ CCL-2266™, neuroblastoma), HeLa (ATCC^®^ CCL-2™, adenocarcinoma of the cervix uterus) and K562 (ATCC^®^ CCL243™, chronic myelogenous leukemia) were maintained in Dulbecco’s modified Eagle’s medium (DMEM for A549, HCT116 and RD); DMEM/F12 1:1 (for SH-SH5Y); Eagle’s minimal essential medium (EMEM for HeLa) and Roswell Park Memorial Institute 1640 medium (RPMI 1640 for K562), with 10% fetal bovine serum, 2-mM l-glutamine and 1% gentamicin as an antibiotic at 37 °C under 5% CO_2_ in a humid atmosphere.

#### 3.3.2. In Vitro Growth Inhibition Assay

The cytotoxicity of the synthesized compounds was determined by the MTT method [7] and Alamar blue test [8]. Cells were seeded at a concentration of 1 × 10^4^ cells/200 µl in a 96-well plate and then incubated (37 °C in a humid atmosphere with 5% CO_2_). After 24 h of incubation, tested compounds were added to the cell cultures at different concentrations from 100 to 1.56 μM, and then, the cells were cultured under the same conditions for 72 h. The effect on cell viability of each concentration was determined in triplicate. All substances were dissolved in DMSO at a final concentration less than 0.1% *v*/*v* (control). For the A549, HCT116, RD, HeLa and HS-SY5Y cell lines, after incubation, 20 μL of MTT (3-(4,5-dimethylthiazol-2-yl)-2,5-diphenyl tetrazolium bromide, 5 mg/mL) was added to each well, and the plates were incubated for a further 2 h. Next, the medium was removed, and 100 μL of DMSO was added to dissolve the formazan crystals formed. The optical density was measured at 536 nm using the Cytation3 (BioTek Instruments, Inc., Winooski, VT 05404, USA) microplate reader. Concentrations (IC_50_) were calculated according to the dose-dependent inhibition curves. The experiments were carried out in triplicate. For K562 cells, after incubation, resazurin (7-hydroxy-3*H*-phenoxazine-3-on-10-oxide sodium salt, 22 μL per one well), from Sigma-Aldrich, with the final concentration of 50 μM was added to each well, and the plates were incubated for another 2 h. The fluorescence of the reduced dye was determined using the Cytation3 microplate reader at excitation at 530 nm and emission at 590 nm. The concentration that caused the 50% inhibition of the growth of the cell population (IC_50_) was determined from the dose-dependent curves.

### 3.4. Enzymatic Assays

#### 3.4.1. Cholinesterases

The ChE inhibition assay was carried out using the Ellman’s spectrophotometric method [23], as applied to the 96-well plate technique [24] on a Tecan Infinite M1000 Pro instrument (Cernusco s.N., Italy). Human recombinant AChE (2770 U/mg) or BChE from human serum (50 U/mg) were incubated in phosphate buffer, pH 8.0, with the tested compounds at different concentrations (typically, seven scalar concentrations ranging from 30 to 0.01 μM) and 5,5′-dithiobis-(2-nitrobenzoic acid) (DTNB) as the chromophoric reagent. After 20 min at room temperature, the substrates acetyl- or butyrylthiocoline were added to the wells, and the increase of absorbance was monitored at 412 nm for 5 min. All the experiments were performed in triplicates (data reported as mean ± SD), and the half-maximal inhibitory concentration (IC_50_) values were calculated by nonlinear regression of the response/concentration (log) curve by using Prism GraphPad software (ver. 5.01).

#### 3.4.2. Monoamine Oxidases

The inhibition of human recombinant MAO A (250 U/mg) and B (59 U/mg; microsomes from baculovirus-infected insect cells) was evaluated using the reported 96-well plate technique [12]. on a Tecan Infinite M1000 Pro instrument (Cernusco s.N., Italy). The test compounds, at different concentrations (typically, seven scalar concentrations ranging from 30 to 0.01 μM), were preincubated 20 min at 37 °C with kynuramine used as the MAO substrate in phosphate buffer at pH 8.0 (0.39 osmolar with KCl). After addition of the enzyme and 30 min of incubation, NaOH was added and the fluorescence read at the 310/400 excitation/emission wavelength. All the experiments were performed in triplicate (data reported as mean ± SD), and IC_50_ values were calculated by nonlinear regression of the response/concentration (log) curve by using Prism GraphPad software (ver. 5.01).

### 3.5. Inhibition Assay of β-Amyloid Aggregation

According to an already reported 96-well plate procedure [25], the test compounds at 100 μM were incubated with Aβ_40_ (30 μM) and 2% HFIP (1,1,1,3,3,3-hexafluoro-2-propanol), used as aggregation enhancer in PBS pH 7.4 for 2 h at room temperature. After the addition of ThT, the fluorescence was read at the 440/485 nm excitation/emission wavelength. Experiments were performed in triplicate, and the % inhibition values calculated as mean ± SD. For compounds showing more than 60% inhibition at a 100-μM concentration, typically, seven scalar concentrations (from 100 to 0.1 μM) of the test compound were evaluated, and IC_50_ values were calculated by nonlinear regression of the response/concentration (log) curve by using Prism GraphPad software (ver. 5.01).

## 4. Conclusions

With this study, we wanted to carry out a further scouting of biological activities associated with the 5,6-dihydro-1-phenylpyrrolo[2,1-*a*]isoquinoline (DHPPIQ) moiety, which is the core structure of type Ia lamellarins. We recently demonstrated the potential of lamellarin-like synthetic compounds, mostly Schiff bases of DHPPIQ 2-aldehydes, as P-gp inhibitors and MDR reversers in a doxorubicin-resistant tumor cell model [4].

The notion that molecules able to evade or inhibit P-gp efflux pumps may cross the BBB and readily distribute into the brain prompted us to synthetically expand the series of the previously reported derivatives with novel aliphatic and aromatic Schiff bases of DHPPIQ 2-carbaldehyde and to investigate their possible interferences with drug targets related to neurological disorders. In this task, we were fully supported by the Multi-fingerprint Similarity Search aLgorithm (MuSSeL), conceived for prioritizing drug targets and suggesting new biological evaluations [5]. Interestingly, in agreement with the MuSSeL predictions, homobivalent Schiff bases assembled on 1,4-phenylenediamine proved to be novel hits for multitarget-directed ligands (MTDLs) addressing Alzheimer’s disease-related target proteins, such as human AChE, MAO and Aβ_40_ aggregation. Among the DHPPIQ Schiff bases, compound **14** proved to be a promising inhibitor of Aβ_40_ self-aggregation (IC_50_ = 13 μM) and AChE (*K*_i_ = 4.69 μM), most likely interacting with the enzymatic PAS. In addition, compound **14** exhibited a moderate inhibition potency toward MAO A, which is a target of antidepressant agents, and a low cytotoxicity.

The lack of effective and long-lasting therapies for AD, due to its multifactorial nature, stimulate medicinal chemists to pursue multitarget drug design strategies. In this context, molecules like **14** might be noteworthy in hit-to-lead optimization studies aimed at developing novel MTDLs, which can slow down the progression of AD, in addition to mitigating its symptoms.

## Data Availability

All data presented in this study are available in the article and in Appendix A.

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
