# Peer review of "Homobivalent Lamellarin-Like Schiff Bases: In Vitro Evaluation of Their Cancer Cell Cytotoxicity and Multitargeting Anti-Alzheimer’s Disease Potential"

_molecules, 2021, doi:10.3390/molecules26020359_

Round 1
Reviewer 1 Report
Nevskaya et al. wrote an interesting article on the use of derivatives of marine alkoloids from the lamellarin family as potential therapeutics against Alzheimer's disease. They used MuSSeL (Multi-fingerprint Similarity Search aLgorithm) to identify DHPPIQs as hits for various targets related to this neurological disease such as: Cholinesterase, Monoamine oxidase, Beta-amiloid aggregation. Based on the similarity search result, the authors designed some non-toxic derivatives of homobivalent paraphenylene DHPPIQ Schiff base with promising low micromolar in vitro activity against AD targets.
The article is written in a scientifically correct way, but it would be nice if the authors wrote more details about the MuSSeL algorithm.
Maybe the authors should also do some molecular docking experiments to get an idea about the receptor-ligand interactions at the atomic level.
I recommend a minor revision.
Author Response
Nevskaya et al. wrote an interesting article on the use of derivatives of marine alkaloids from the lamellarin family as potential therapeutics against Alzheimer's disease. They used MuSSeL (Multi-fingerprint Similarity Search aLgorithm) to identify DHPPIQs as hits for various targets related to this neurological disease such as: Cholinesterase, Monoamine oxidase, Beta-amyloid aggregation. Based on the similarity search result, the authors designed some non-toxic derivatives of homobivalent paraphenylene DHPPIQ Schiff base with promising low micromolar in vitro activity against AD targets.
R: Thanks for the favorable comments.
The article is written in a scientifically correct way, but it would be nice if the authors wrote more details about the MuSSeL algorithm.
R: Further details about MuSSeL and related references have been added in the paragraph “3.2. Chemoinformatics and computational chemistry” (see lines 511-520).
Maybe the authors should also do some molecular docking experiments to get an idea about the receptor-ligand interactions at the atomic level.
R: We believe that it is not worth at this stage of our research. In our opinion, docking calculation would not add more information than that obtained by the similarity searching algorithm, that is MuSSeL in this case, which identified several structurally similar ligands and, of course, similar binding modes to the related protein targets (see lines 190-192). Molecular docking calculation is generally acknowledged to be more effective in lead optimization studies (and indeed we’ll carry out such a calculation!), while it could return a high number of false positives in earlier stages. The Schiff bases of the lamellarin-like synthetic aldehydes, whose probable AD-related target proteins have been prioritized through MuSSeL, are at most hit/lead compounds, with IC50 values in the low μM range.
I recommend a minor revision.
Reviewer 2 Report
In this paper, the authors present an efficient synthetic approach for the preparation of imino adducts of 5,6-dihydro-1-phenylpyrrolo[2,1-a]isoquinoline (DHPPIQ) 2-carbaldehyde by the condensation reaction between the respective heteroarylaldehyde with aliphatic or aromatic amines. Additionally, these Schiff bases were successfully evaluated for their cytotoxicity in five diverse tumor cell lines. In general, the thematic and results are interesting for the chemists of the synthetic-organic and medicinal field as well as for they who investigate in drug discovery. Likewise, the manuscript is well presented, the methods are very convenient and practical, the synthesized products were well characterized (by IR, NMR, MS, and elemental analysis), and the biological results are complete and conclusive. Therefore, I believe this paper might be appropriate for its publication in Molecules journal after some (minor) revisions and answers for the questions and suggestions:
- Keywords must be in alphabetical order.
- In general, authors should carry out a better and complete analysis for section 2.1. Chemistry. They must also place a more pertinent conclusion in this regard.
- Include the time (or time range) of reaction in the note of Scheme 1. Also, include the yield (or yield range) of the respective reactions in the respective structures.
- Materials and Methods. Place compound names in the same font format (normal or italic).
- In SI, Include at least NMR spectra copies for synthesized imines.
- Place 'in vitro' in italic wherever it appears.
- The letter 'J' of coupling constants must be italicized.
- ETC.
Author Response
In this paper, the authors present an efficient synthetic approach for the preparation of imino adducts of 5,6-dihydro-1-phenylpyrrolo[2,1-a]isoquinoline (DHPPIQ) 2-carbaldehyde by the condensation reaction between the respective heteroarylaldehyde with aliphatic or aromatic amines. Additionally, these Schiff bases were successfully evaluated for their cytotoxicity in five diverse tumor cell lines. In general, the thematic and results are interesting for the chemists of the synthetic-organic and medicinal field as well as for they who investigate in drug discovery. Likewise, the manuscript is well presented, the methods are very convenient and practical, the synthesized products were well characterized (by IR, NMR, MS, and elemental analysis), and the biological results are complete and conclusive. Therefore, I believe this paper might be appropriate for its publication in Molecules journal after some (minor) revisions and answers for the questions and suggestions:
- Keywords must be in alphabetical order.
R: Keywords have been re-ordered (lines 36-37)
- In general, authors should carry out a better and complete analysis for section 2.1. Chemistry. They must also place a more pertinent conclusion in this regard.
R: The reaction conditions were optimized for improving the synthetic effectiveness as demonstrated by the satisfactory yields achieved. A few changes have been done in the paragraph 2.1 (lines 75-103).
- Include the time (or time range) of reaction in the note of Scheme 1. Also, include the yield (or yield range) of the respective reactions in the respective structures.
R: Reaction times and yields (sometimes ranges) have been complemented in the caption to Scheme 1 (lines 121-124).
- Materials and Methods. Place compound names in the same font format (normal or italic).
R: All the compound names are now italicized.
- In SI, include at least NMR spectra copies for synthesized imines.
R: All the 1H and 13C NMR spectra of the newly synthesized imino adducts have been included in Supp. Info. file.
- Place 'in vitro' in italic wherever it appears.
R: Done
- The letter 'J' of coupling constants must be italicized.
R: Done
- ETC.
R: The manuscript text has been revised in several parts.
Reviewer 3 Report
The manuscript entitled "Homobivalent lamellarin-like Schiff bases as prospective multitarget anti-Alzheimer’s disease agents" by the group of Cosimo
D. Altomare describes the potential multimodal activity of some 5,6-dihydro-1-
21 phenylpyrrolo[2,1-a]isoquinoline-bearing Schiff bases.
1. My main objection is about the cytotoxicity results which I believe do not provide any further evidence on the main target of this research which seems to the multimodal activity of the compounds (hence the title). I feel that this section should be removed.
2. The synthetic part of the paper is presented as a convergent kind of synthetic approach. There is no description of 1 and 2. I would suggest that the synthesis of 16, 17, 18,19 should be part of a second scheme without the generalised structures and describe a table with the structure of 1-15.
3. There is no discussion on the structure activity relationship except from the planarity and the Abeta activity. PLease expend a bit more in general in the text in addition to simply stating the results.
4. Line 198 "showed nil or poor selectivity toward MAO isoforms". First of all it would help if you add a column in the table stating the selectivity index and what you expected to be a good or excellent selectivity. For compound 14 for example, 12 micromolar caused 50% inhibition for MAO A and 10 micromolar caused only 19% inhibition. Is the 2.5X good or bad??? your positive control is almost 5x more selective. Please comment.
5. Please replace all the "IC50s" with "IC50 values"
6. PLease add NMRs in the SI
7. Methods and materials: please put all compound names in italics
8. Please replace the titles "enzymes' inhibition", "Monoamine oxidases’ inhibition", "Cholinesterases’ inhibition"
9. toooooo long sentences. The introduction is 5 sentences. Please use shorter sentences.
10. For the ThT test did you check the optical properties of the compounds such as excitation/emission. I would suggest that for the most active compound 14, circular dichroism experiments should be employed to confirm your results.
Author Response
The manuscript entitled "Homobivalent lamellarin-like Schiff bases as prospective multitarget anti-Alzheimer’s disease agents" by the group of Cosimo D. Altomare describes the potential multimodal activity of some 5,6-dihydro-1-phenylpyrrolo[2,1-a]isoquinoline-bearing Schiff bases.
- My main objection is about the cytotoxicity results which I believe do not provide any further evidence on the main target of this research which seems to the multimodal activity of the compounds (hence the title). I feel that this section should be removed.
R: We would like to maintain this section for two main reasons: (i) It is a conjunction with previous works of ourselves (reff. 3 and 4) and others (reff. 1 and 2), focused on antiproliferative/cytotoxic activity of lamellarin-like fused azaheterocycles; (ii) the data in Table 1 provide useful assessment of the cytotoxicity of novel compounds proposed for other pharmacological targets.
- The synthetic part of the paper is presented as a convergent kind of synthetic approach. There is no description of 1 and 2. I would suggest that the synthesis of 16, 17, 18, 19 should be part of a second scheme without the generalized structures and describe a table with the structure of 1-15.
R: We have slightly modified Scheme 1 and incorporated a small table with R1-R4 substituents on the aldehyde precursors 1-6. The cytotoxicity data of compounds 1 and 2, taken from a previous work from the same laboratory, have been included in Table 1. With these changes, we may maintain one synthesis scheme.
- There is no discussion on the structure-activity relationship except from the planarity and the Abeta activity. Please, expend a bit more in general in the text in addition to simply stating the results.
R: The paragraphs related to SARs have been extended to shed light on points not enough discussed in the initial manuscript version (see lines 215-233).
- Line 198 "showed nil or poor selectivity toward MAO isoforms". First of all, it would help if you add a column in the table stating the selectivity index and what you expected to be a good or excellent selectivity. For compound 14 for example, 12 micromolar caused 50% inhibition for MAO A and 10 micromolar caused only 19% inhibition. Is the 2.5X good or bad??? your positive control is almost 5x more selective. Please comment.
R: The text has been revised (see now lines 264-265), hopefully avoiding risks of confusion. We think that the addition of a further column with selectivity ratios is not necessary. In any case, the isoform selectivity of our compounds is quite poor and lower than that of pargyline.
- Please replace all the "IC50s" with "IC50 values"
R: All the suggested changes have been done.
- Please add NMRs in the SI
R: All the 1H and 13C NMR spectra of the newly synthesized Schiff bases have been included in Supp. Info. file.
- Methods and materials: please put all compound names in italics
R: All the compound names are now italicized.
- Please replace the titles "enzymes' inhibition", "Monoamine oxidases’ inhibition", "Cholinesterases’ inhibition"
R: The titles of these paragraphs in the experimental section have been changed (lines 550, 551 and 564).
- toooooo long sentences. The introduction is 5 sentences. Please use shorter sentences.
R: The introductory text has been shortened, some repetitions have been deleted and long sentences have been split.
- For the ThT test did you check the optical properties of the compounds such as excitation/emission. I would suggest that for the most active compound 14, circular dichroism experiments should be employed to confirm your results.
R: Thanks for the suggestion. We’ll extend the biophysical and spectroscopic in the optimization stage of compound 14.
Round 2
Reviewer 3 Report
The authors have appropriately addressed almost all the points previously mentioned. Still I would like to express my concern about the persistence to include the first part of the results related to the cancer cell cytotoxicity in a manuscript entitled "Homobivalent lamellarin-like Schiff bases as prospective multitarget anti-Alzheimer’s disease agents". There is no other mentioning of the anticancer-activity, not in the key-words, not in the title.
The reviewer appreciates completely the contribution of the results of Table 1 and the reasoning provided and since the authors feel so strongly about keeping this set of results then a title change is necessary to something like "Homobivalent lamellarin-like Schiff bases: in vitro evaluation of cancer cell cytotoxicity and their multitargeting anti-Alzheimer’s disease potential."
Line 118. : Please change the title "2.3. Protein targets’ prediction by similarity searching" to "target protein prediction by Similarity Search"
Author Response
Replies in bold
The authors have appropriately addressed almost all the points previously mentioned. Still I would like to express my concern about the persistence to include the first part of the results related to the cancer cell cytotoxicity in a manuscript entitled "Homobivalent lamellarin-like Schiff bases as prospective multitarget anti-Alzheimer’s disease agents". There is no other mentioning of the anticancer activity, not in the keywords, not in the title.
R: We propose to add “cytotoxicity” among the keywords (line 37).
The reviewer appreciates completely the contribution of the results of Table 1 and the reasoning provided and since the authors feel so strongly about keeping this set of results then a title change is necessary to something like "Homobivalent lamellarin-like Schiff bases: in vitro evaluation of cancer cell cytotoxicity and their multitargeting anti-Alzheimer’s disease potential."
R: We thanks the reviewer for her/his good suggestion and accordingly modified the manuscript title.
Line 118: Please change the title "2.3. Protein targets’ prediction by similarity searching" to "target protein prediction by Similarity Search"
R: We have changed also this title.